# Effect of Low Pressure and Low Oxygen Treatments on Fruit Quality and the In Vivo Growth of *Penicillium digitatum* and *Penicillium italicum* in Oranges

John Archer [1,2,*], Penta Pristijono [1], Quan V. Vuong [1], Lluís Palou [3] and John B. Golding [1,2]

1 School of Environmental and Life Sciences, University of Newcastle, Ourimbah, NSW 2258, Australia; penta.pristijono@newcastle.edu.au (P.P.); vanquan.vuong@newcastle.edu.au (Q.V.V.); john.golding@dpi.nsw.gov.au (J.B.G.)
2 NSW Department of Primary Industries, Ourimbah, NSW 2258, Australia
3 Pathology Laboratory, Postharvest Technology Center (CTP), Valencian Institute of Agrarian Research (IVIA), 46113 Montcada Valencia, Spain; palou_llu@gva.es
* Correspondence: john.archer@uon.edu.au

**Abstract:** *Penicillium digitatum* and *P. italicum* are the major postharvest pathogens in citrus. To reduce postharvest decay, the use of low-oxygen (0.9 kPa $O_2$) (LO) or low-pressure (6.6 kPa) (LP) treatments were evaluated during the storage of navel oranges for four or eight days. The results showed that exposure to both LO and LP treatments reduced in vivo pathogen growth compared to the untreated (UTC) oranges, with LO being the most effective. The effects of LO and LP on fruit metabolism and quality were further assessed, and it was found that there was no effect on fruit ethylene production, respiration rate, TSS (total soluble solids), TA (titratable acidity) or fruit firmness. However, both LO and LP treatments did have an effect on juice ethanol concentration and fruit weight-loss. The effect of adding exogenous ethylene at either LP (1 μL/L) or atmospheric pressure (AP) (at either 0.1, 1 μL/L) was also evaluated, and results showed that the addition of ethylene at these concentrations had no effect on mould diameter at LP or AP. Therefore, both LO of 0.9 kPa $O_2$ and LP of 6.6 kPa at 20 °C are potential non-chemical postharvest treatments to reduce mould development during storage with minimal effects on fruit quality.

**Keywords:** citrus; *Citrus sinensis*; low pressure; low oxygen; green mould; blue mould

## 1. Introduction

Oranges are important fruit which are widely grown and traded around the world. The optimum storage temperature for oranges is 2 to 3 °C at 85% to 90% relative humidity (RH), allowing for storage times of between 4 to 8 weeks [1]. However, postharvest decay is the major problem for the storage of citrus fruit. *Penicillium digitatum* (Pers.: Fr.) Sacc. and *Penicillium italicum* Wehmer are responsible for green and blue mould, respectively, in citrus and are the major postharvest pathogens which cause significant commercial costs [2]. Infection of both *P. digitatum* and *P. italicum* in citrus fruits starts with a rind wound, from which a lesion develops by the softening of the fruit tissue and maceration of the cells caused by pectinases secreted by the fungus [3]. Both fungi are wound pathogens, requiring a microwound in the skin to allow infection. These pathogens are currently controlled with commercial synthetic fungicides such as imazalil. However, consumer and regulatory concerns have resulted in renewed interest in environmentally friendly and safe alternatives to the use of postharvest fungicides [4]. Such non-chemical treatments include physical treatments of low pressure (LP) and low oxygen (LO), which may have some potential to reduce postharvest decay in the storage of many fruits and vegetables [5,6], and may have some benefit for the storage of citrus.

LP storage technology is not a new technology but has been more recently used as a storage technique that can rapidly remove heat, reduce oxygen levels, and rapidly remove

and manage the storage atmosphere (Wang et al., 2001). Previous studies show the promise of combined LO with LP treatments, sub-atmospheric pressure, and cold storage [7–9]. LP treatments have been shown to improve storage outcomes by reducing oxygen, other volatiles and ethylene levels around citrus fruit [10]. In lime fruits, LP storage at 22.6 kPa was shown to maintain peel colour and flavour during storage [11]. It has been reported that LP treatments maintained the quality of fresh strawberries due to the reduction of oxygen pressures, reducing storage decay [12]. Other researchers have suggested the benefits of LP are facilitated by the removal/reduction of volatile gases, such as ethylene and $\alpha$-farnesene, or the stimulation of an 'immune' response in the fruit itself [13–15]. Several studies have reported that LP lowers postharvest decay and postharvest browning in a range of horticultural crops. For example, tomatoes stored at LP 4 kPa at 10 °C demonstrated significantly decreased water loss, with a reduction of calyx browning of 12% and calyx rots of 16% [16]. Furthermore, green capsicum (pepper) stem decay was shown to be lower when stored at 4 kPa compared to that of regular atmospheric storage of 101 kPa [17]. However, there are limited studies on the effect of LP treatments on citrus fruits and their effect on citrus-decaying fungi such as *P. digitatum* and *P. italicum*.

There is an inherent physical interaction of LP and LO wherein decreasing atmospheric pressure reduces oxygen pressure. For example, at normal atmosphere (101 kPa) the oxygen level is 20.95% with a partial pressure of 20.7 kPa, but at lower pressures, such as 6.6 kPa, the level of oxygen in the air is equivalent to 13.6%, with a partial pressure of 0.9 kPa. This change in partial pressures is due to the increased percentage of water ($H_2O$) which is required to maintain 100% humidity at that pressure and at the same set temperature. To maintain 100% humidity within the treatment chamber, the partial pressure of water must remain the same irrespective of pressure, as it is temperature dependent. This inherent decrease in oxygen in the air at low pressures should be considered when studying low-pressure treatments [5]. LO treatments have been shown to reduce the in vitro growth of fungal pathogens causing postharvest decay in a range of horticultural produce [5]. Oxygen ($O_2$) levels below 1% have been found to reduce mould growth, but these levels do not kill the fungus, allowing the vegetative fungus to continue growing once oxygen returns to normal levels [18]. Similar observations were made when mandarins and oranges artificially inoculated with *P. digitatum* or *P. italicum* were exposed to atmospheres of 95 kPa $N_2$, 95 kPa $CO_2$ or 30 kPa $O_2$ [19].

The aim of this study was to compare the effects of LO and LP treatments on the quality of orange fruit and the growth of *P. digitatum* and *P. italicum* on artificially inoculated navel oranges incubated at 20 °C.

## 2. Materials and Methods

Four different experiments were conducted. Experiment 1, 2 and 4 were designed to assess the effect of the treatments on disease development on oranges, whereas the effect on orange quality was studied in Experiment 3.

### 2.1. Fruits and Inoculation

Fungicide-free navel oranges (*Citrus sinensis* L. Osbeck, cv. Bellamy) were harvested from the Somersby Research Station farm at the NSW Department of Primary Industries (NSW DPI), Australia and used for decay assessments. Blemish-free fruits were harvested at commercial maturity (total soluble solids (TSS) 13.7 °Brix, titratable acidity (TA) 0.85% citric acid and maturity index (MI = TSS/TA ratio) of 16.1). After harvest, fruits were washed, surface disinfected with 50 µL/L free chlorine water at room temperature (20 °C) and allowed to air dry. Fruits were used the same day as harvest or stored up to 7 days at 5 °C and 90% relative humidity (RH).

For Experiment 3, fungicide-free Valencia oranges (*Citrus sinensis* L. Osbeck, cv. Valencia Late) at commercial maturity were harvested from a commercial orchard at Yanco, NSW, Australia.

Fruits for Experiment 1 were inoculated with *P. italicum* CAN-23 spores, while fruits for Experiment 2 and 4 were inoculated with both *P. digitatum* CAN67 and *P. italicum* spores on opposite sides of the fruit. Wild type *P. digitatum* and *P. italicum* spores ($5.2 \times 10^6$ spores per mL) obtained from the NSW DPI citrus pathology laboratory were inoculated onto a wound in each orange by dipping a stainless steel rod into the inoculum solution, and then immediately making a puncture 2 mm deep in the fruit [20]. Inoculated fruits were then incubated for 24 h at 20 °C before treatment started. After treatment, the infection rates (disease incidence, %) and the growth of *Penicillium* in navel oranges (disease severity, lesion diameter, mm) were assessed 3 times after removal at 24 h, 4 days and 7 days after treatment. The average of two perpendicular directions of the fungal lesion on the fruit was used for the statistical analysis.

Experiment 1 assessed LO (0.9 kPa) and LP (6.6 kPa) using 270 oranges divided into 18 trays of 15 fruits. Experiment 2 assessed LO (0.9 kPa) and LP (6.6 kPa) using 540 oranges divided into 18 lots of 30 fruits. Each treatment was independently replicated 3 times in different bags, chambers, or drums. Experiment 3 assessed LO (0.9 kPa) and LP (6.6 kPa) using 1140 oranges divided into 76 trays of 15 fruits. Experiment 4 assessed LP (6.6 kPa with and without 1 μL/L exogenous ethylene) and atmospheric pressure (at either 0.1 or 1 μL/L exogenous ethylene) using 225 oranges divided into 15 trays of 15 fruits. A summary of the concentrations and partial pressures of the main vapours in each treatment, including the untreated control (UTC), LP, and LO treatments, are presented in Table 1, with the relative humidity (RH) maintained at 100% in all cases.

**Table 1.** Summary of the gas pressures and equivalent percentage of gasses of each treatment used in experiments at 20 °C. UTC = untreated control, LP = low pressure and LO = low oxygen.

| | UTC 101.3 kPa + 100% RH | | LP 6.6 kPa + 100% RH | | LO 101.3 kPa + low $O_2$ + 100% RH | |
|---|---|---|---|---|---|---|
| | % | kPa | % | kPa | % | kPa |
| | | 101.3 | | 6.6 | | 101.3 |
| $H_2O$ | 2.30% | 2.3 | 35.00% | 2.3 | 2.3% | 2.3 |
| $N_2$ | 76.3% | 77.3 | 50.8% | 3.4 | 96.7% | 98.0 |
| $O_2$ | 20.5% | 20.7 | 13.6% | 0.9 | 0.9% | 0.9 |
| Ar | 0.9% | 0.9 | 0.6% | 0.04 | 0.04% | 0.04 |
| $CO_2$ | 0.04% | 0.04 | 0.0004% | 0.00172 | 0.02% | 0.02 |
| Total | 100% | 98.932 | 100% | 4.3 | 100% | 98.99 |

### 2.2. LO Storage Treatments

LO storage treatments were conducted in 60-L steel drums that were used to seal the fruit and that contained a modified atmosphere of 1% $O_2$ in a flow-through system at 200 mL/min (equivalent to 1 air exchange every 4 h). The levels of oxygen within the flow-through system were regularly assessed using a paramagnetic sensor (Servomex Paramagnetic $O_2$ Transducer Series 1100). To maintain 100% RH within the drum, the modified atmosphere was passed through a humidifier before entering the treatment drums.

### 2.3. LP Storage Treatments

A (VivaFresh) low-pressure system consisting of 6 identical aluminium chambers (0.61 L × 0.43 W × 0.58 H m total volume of 0.152 m$^2$) was used to maintain the LP treatments. The vacuum was produced and maintained with a two-stage rotary vacuum pump (Model 2005I, Pfeiffer Vacuum SAS (former Adixen), Nashua, NH, USA) which used a solenoid valve controlled by a proportional/integral/derivative (PID) computer control system using VivaFresh software and control parameters. The low-pressure system was able to alter an air flow controller to adjust the air exchange rate to ensure that accumulation of metabolic gasses did not occur, as all air within the chamber was exchanged every 2 h. Air was humidified with a humidifier before entering the chamber, and the internal air humidity levels were monitored with wet-bulb and dry-bulb temperatures using calibrated

thermistors (YSI 55,000 Series GEM). The data created by the different sensors was sent to the control box and then accessed via ethernet by computers on the network [17,21].

### 2.4. Untreated Control

The untreated control fruit were stored in an unsealed but folded-over LDPE bag at 20 °C. The RH within the bag was above 95%, which was monitored using calibrated temperature and relative humidity sensors (Tinytag TV-4500 data loggers, Gemini, Chichester, UK).

### 2.5. Fruit Quality

The respiration rate (as evolved $CO_2$) was measured from a 5 mL sample of the headspace from a sealed 1500 mL septum-containing respiration glass jar housing the fruit for 4 h. The respiration rate of the fruits was calculated to within 0.1% using an ICA40 series low-volume gas analysis system (International Controlled Atmosphere Ltd., Kent, UK). Respiration rate was determined using the below formula:

$$R(CO_2) = \frac{(CO_2(\%) \times \text{volume (mL)})}{(\text{weight(kg)} \times \text{time(h)} \times 100)}$$

where, $R(CO_2)$ is respiration rate and is expressed as mL $CO_2$/kg/h [22].

Ethylene concentration was measured by withdrawing a 1 mL sample from the headspace from a 1500 mL septum-containing respiration glass jar housing the fruit for 4 h and then injecting into a flame ionisation gas chromatograph (Gow-Mac 580, Bethlehem, PA, USA) fitted with a ($6' \times 1/8''$) activated alumina stainless steel carbowax silico steel 80/100 column. The operating temperature for the detector was 105 °C; for the column, the operating temperature was 85 °C and the operating temperature for the injectors was 65 °C. Nitrogen was used as the carrier gas at a flow rate of 60 mL/min [23,24].

Ethylene production was calculated as:

$$R(C_2H_4) = \frac{\left(C_2H_4\left(\mu L\ L^{-1}\right) \times \text{volume(mL)}\right)}{(\text{weight(kg)} \times \text{time(h)})}$$

where $R(C_2H_4)$ is rate of ethylene production and is expressed as $\mu L\ C_2H_4$ /kg/h [22].

The TSS of each fruit was determined using a portable digital refractometer (Atago, Saitama, Tokyo, JP) and expressed as °Brix. Titratable acidity (TA) was measured by titrating 5 mL of the combined juice of 10 fruits with 0.1 N NaOH to pH 8.2 by an automatic titrator (Mettler Toledo, Greifensee, CH) and was expressed as percentage citric acid. To determine the ethanol content of the fruit, 10 mL of freshly squeezed juice from the combined juice of 10 fruits was transferred into a glass vial (20 mL) with crimp-top caps sealed with a silicone septum and incubated in a water bath at 30 °C for 10 min. A headspace sample (1 mL) was drawn from the vials and analysed for ethanol by gas chromatography (Model 580, Gow-Mac, Bethlehem, PA, USA) equipped with a flame ionization detector and a Carbowax column (Gow-Mac, Bethlehem, PA, USA).

The detector operating temperature was 190 °C, the column was 68 °C and the injector was 190 °C. The gas flow rates were 300 mL/min for air, 30 mL/min for hydrogen and 30 mL/min for nitrogen, respectively. The concentration of ethanol was expressed as µL/L [23].

### 2.6. Ethylene Treatment

Ethylene from a standard cylinder (BOC Gases, Sydney, NSW, Australia) was mixed with air to generate gas streams of 1 or 0.1 µL/L ethylene, which was monitored with a GC (as previously described). The humidity was maintained at 100% RH by passing the modified atmosphere through a humidifier before entering the treatment barrels at 200 mL/min. A gas treatment line was set up to supply the low-pressure system, thus enabling ethylene intake into the low-pressure system.

### 2.7. Weight Loss

Fruit weight loss was expressed as grams comparing the initial weight with the weight at the assessment time using a calibrated NavigatorXT (OHAUS, Parsippany, NJ, USA) balance.

### 2.8. Fruit Firmness

Fruit firmness was assessed with a texture analyser (Lloyd Instrument LTD, Fareham, UK) to determine the firmness of 10 fruits per replicate at a controlled room and fruit temperature of 20 °C. The fruit was assessed by compressing in the equatorial zone between two flat surfaces closing together at a speed of 1 mm/min, and the fruit firmness (J) was recorded.

### 2.9. Statistical Analysis

A randomized factorial experiment design with three factors (LP, LO and duration of treatment) was used with three independent replicates, where each replicate was separately treated in a different LP chamber and LO drum. For Experiments 1 and 4, the experimental unit was 15 fruits, while it was 30 fruits for Experiment 2. Experiment 3 used 10 individual fruits for firmness and TSS. To assess for ethylene and $R(CO_2)$, 2 fruits were sealed in a respiration jar and one gas sample was evaluated. Ethanol and TA were assessed by combining the juice of 10 fruits. A one-way ANOVA was used to determine the level of difference between different treatments and treatment durations. Tukey honestly significant difference (HSD) test ($p < 0.05$) was completed with JMP Pro 14.2.0, and Student's T-tests and least significant differences (LSD, $p < 0.05$) were also completed with JMP Pro 14.2.0.

## 3. Results and Discussion

The effect of LO and LP in Experiment 1 is presented in Figure 1. The results show a significant difference between the treatments in their effect on the growth of *P. italicum* in inoculated oranges. Blue mould severity in UTC fruit was higher than that in fruit subjected to LO and LP treatments, except after 7 days, where there was no difference between the LP 4-day treatment and the UTC average. The growth of *P. italicum* in oranges subjected to the LO treatment was found to be lower than in UTC oranges. The infection rate of *P. italicum* in this experiment was above 95% and was not affected by treatment.

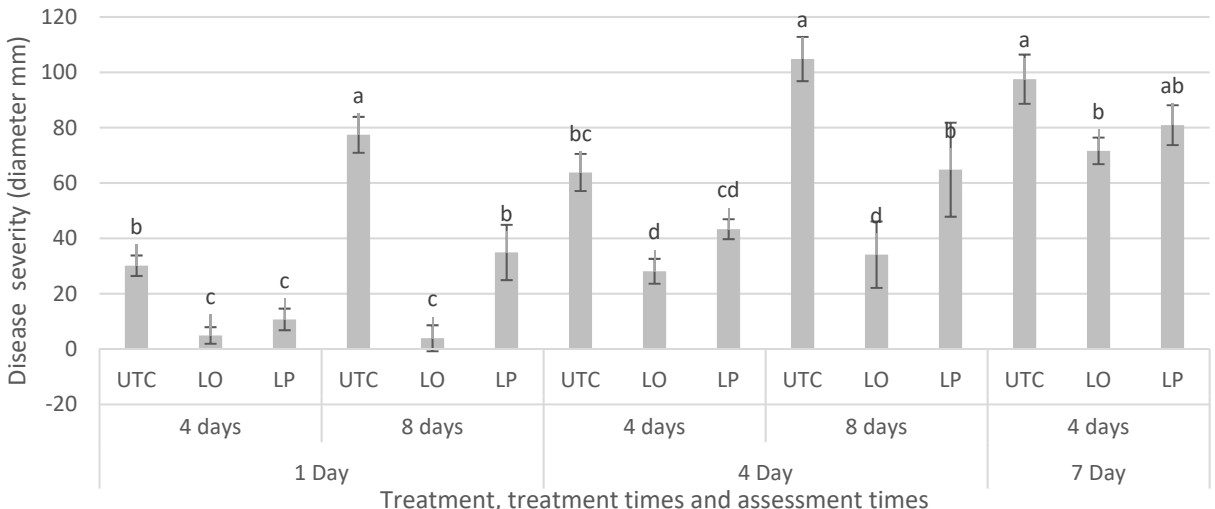

**Figure 1.** Severity of blue mould (mm) caused by *Penicillium italicum* in untreated fruit (UTC) and fruit exposed to low oxygen (LO) and low pressure (LP) following two treatment durations (4 and 8 days) and assessed after an additional period of 1, 4 or 7 days after treatment. The values are the means of 3 replicates of 15 oranges. For each assessment time, different letters denote significant differences between treatments. Vertical lines above columns designate standard deviation. Each assessment time was analysed independently with an alpha of 0.05, 24 h LSD of 15.7, 4 days LSD of 20.7 and 7 days LSD 20.1.

Within the LO treatments, after treatment and storage in air for 4 days, there was low mould growth and no significant difference in blue mould severity between the LO 4-day treatment and LO 8-day treatment. These results are in agreement with previous observations, where strawberry and raspberry fruits stored in a low-oxygen environment (3–5 kPa $O_2$ and 5–10 kPa $CO_2$) had low mould development up to 14 days of treatment [25].

In this study, there was a significant decrease of *P. italicum* growth in oranges following LP and LO treatments compared to UTC. Observations highlighted in Figure 2 show the UTC fruit having large amounts of visible decay and growth of *P. italicum*, which was sporulating after 8 days of storage (Figure 2a), as compared to LP results, which were observed to be less and as expected [26]. LO-treated oranges showed the least amount of visible decay and mould development (Figure 2c), with the least number of visible spores, whilst the LP-treated oranges showed reduced amounts of decay compared to UTC (Figure 2b).

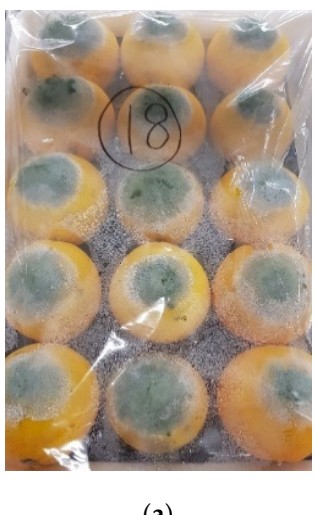 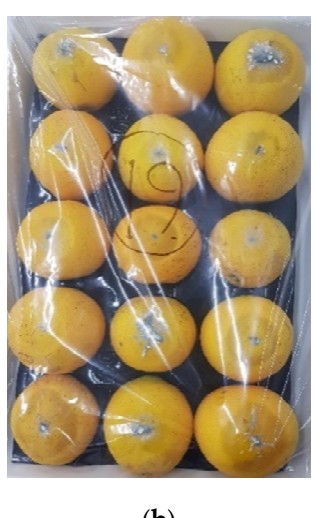 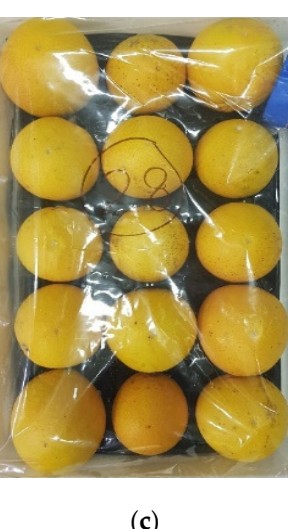

(**a**) (**b**) (**c**)

**Figure 2.** Growth and sporulation of *Penicillium italicum* in infected oranges after 8 days of treatment for UTC fruits ((**a**)—left), LP-treated fruits, ((**b**)—middle) and LO-treated fruits ((**c**)—right), plus an additional 24 h in air following treatment.

The results show that both the LO and LP treatments were able to reduce the disease severity of *P. italicum* in infected oranges. These results align with the previous findings that LP treatment resulted in lower decay development in both zucchini [27] and tomato fruit [17], while LO treatment was able to reduce mould development in strawberries [18,25].

Although the partial pressure of $O_2$ in both the LO and LP treatments was maintained at the same $O_2$ partial pressure (i.e., 0.9 kPa in both treatments), the results showed that LO treatment resulted in a lower growth rate of *P. italicum* in infected oranges for the 1-day assessment (Figure 1). These results were further examined in Experiment 2, which assessed a treatment time of 4 days for both blue and green moulds (Figure 3). These results show that, although there were differences in decay development rate detected 1 day after treatment, blue mould only showed a difference between LO when compared to either UTC or LP. When comparing disease incidence (infection rate), there was no treatment or treatment duration effect (data not shown). LO was more effective than LP in lowering disease severity (Figures 1 and 3), and it may be that in the LP, the $CO_2$ diffusion was higher within the fruit and storage atmosphere, which may have augmented the potential antifungal effect [5,18]. LP treatments also facilitated a higher mobility of the metabolic gases, which may also have an impact on fungal growth. Due to the nature of these treatments, the air exchange in LP treatments were at a higher rate than that of the LO treatment, so it might be the metabolic gases that caused the LO treatment to induce

lower mould development. In general, it has been shown that low $O_2$ levels below 1% $O_2$ may develop an 'off-taste' in produce and do not kill bacteria or mould [5].

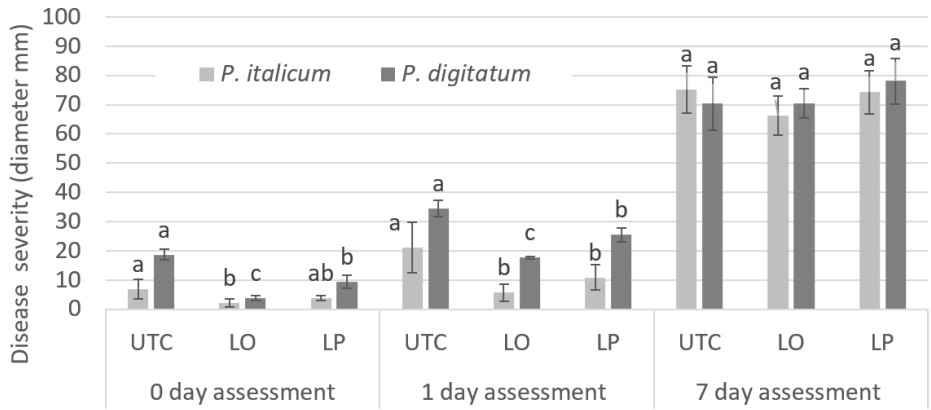

**Figure 3.** Disease severity (mm) of blue mould caused by *Penicillium italicum* and green mould caused by *Penicillium digitatum* in oranges which were untreated (UTC) or treated with low oxygen (LO) or low pressure (LP) for 4 days and assessed 0, 1, or 7 days after treatment. The values are the means of 3 replicates of 30 oranges. Each assessment time was analysed independently with Student's t-test and an alpha of 0.05. Mean values with different letters indicate significant differences among pathogens. Vertical lines above columns designate standard deviation. *P. italicum* had an LSD 4.25 for the 0 h assessment, 8.7043 for the 24 h assessment and 17.138 for the 7 day assessment. *P. digitatum* had an LSD of 5.20 for the 0 h assessment, 5.985 for the 24 h assessment and 19.60 for the 7 day assessment.

The effect LP and LO had on citrus fruit quality was also examined. The results showed that after 4 or 8 days of treatment and up to 7 days shelf life at 20 °C, there was no effect on fruit TSS, TA, or in fruit respiration rates, ethylene production or fruit firmness (Table 2, Figure S1). However, fruit treated with both LP and LO treatments had significantly higher ethanol levels than the control fruit, such that the longer LP and LO treatment times (8 days) resulted in higher ethanol concentrations as compared to the shorter LP and LO (4 days) treatments. This suggests that with extended use of either LP or LO treatments, ethanol levels may build up to concentrations that would become unpalatable for consumers. Hagenmaier [28] found that an ethanol concentration of >1500 µL/L was required for consumers to taste ethanol flavours in citrus. However, the LP and LO (8 day) treatments were significantly below this ethanol level, with the highest reading at 94 µL/L (Table 2).

Both the LO and LP (4 and 8 day) treatments also reduced weight loss in comparison to the UTC (4 and 8 day) treatments. At 8 days, the UTC demonstrated a weight loss of 102.2 g, which is as compared to the weight loss of LP- and LO-treated fruits at 24.2 g and 20.7 g, respectively. With this reduced weight loss benefit, the weight of the product may be maintained for longer and improve profitability.

The addition of ethylene (1 µL/L) to the storage atmosphere in the LP treatment had no effect on disease severity. Indeed, the addition of any concentration of exogenous ethylene (1 and 0.1 µL/L) in the storage atmosphere had no effect on lesion size (Figure 4). All fruit subjected to LP treatments showed significantly lower severity of both green and blue moulds compared to all atmospheric pressure treatments (air, 0.1 µL/L ethylene and 1 µL/L ethylene) (Figure 4). This is in agreement with previous results that showed no role for ethylene in the in vitro growth of *P. digitatum* or *P. italicum* or in the incidence of green and blue moulds on artificially inoculated citrus fruits [29]. However, green and blue mould severity increased on mature oranges and mandarins exposed to de-greening with

exogenous ethylene (2 µL/L ethylene at 21 °C and 95–100% RH for 3 days), which was attributed to an aging effect (increased senescence) on the fruit rind [29].

**Table 2.** Effect of low oxygen (LO) and low pressure (LP) on the quality of Valencia oranges stored for 0, 4 and 7 days at 20 °C.

| | | Ethylene Production (µL $C_2H_4$/kg/h) | Respiration Rate (mL$CO_2$/kg/h) | Ethanol (µL/L) | TSS (°Brix) | TA (% Citric Acid) | MI | Weight Loss (g) | Firmness (J) |
|---|---|---|---|---|---|---|---|---|---|
| | Treatment duration | | | | | | | | |
| | Upon removal from treatment | | | | | | | | |
| Time Zero | 0 | 9.7 | 8.1 | 4.2 [b] | 9.7 | 1.0 | 9.3 | 0.0 [d] | |
| UTC | 4 | ND | 5.4 | 4.9 [b] | 9.8 | 1.1 | 9.2 | 47.6 [b] | 0.05 |
| | 8 | ND | 6.2 | 6.6 [b] | 9.6 | 1.1 | 9.1 | 102.2 [a] | |
| Low Oxygen | 4 | ND | 5.4 | 27.2 [b] | 9.5 | 1.0 | 9.6 | 14.9 [c] | 0.05 |
| | 8 | ND | 6.2 | 53.9 [a] | 9.5 | 1.0 | 9.7 | 20.7 [c] | |
| Low Pressure | 4 | ND | 5.4 | 61.8 [a] | 9.5 | 1.0 | 9.8 | 24.9 [c] | 0.06 |
| | 8 | ND | 6.2 | 65.9 [a] | 9.5 | 1.1 | 8.9 | 24.2 [c] | |
| LSD | | NS | 24.0 | NS | NS | NS | NS | 11.5 | NS |
| | After 4 days air storage at 20 °C | | | | | | | | |
| UTC | 4 | ND | 7.4 | 4.1 [d] | 9.6 | 1.0 | 9.4 | 99.9 [a] | |
| | 8 | ND | 3.9 | 3.6 [d] | 9.6 | 1.1 | 8.9 | 111.8 [a] | 0.03 |
| Low Oxygen | 4 | ND | 6.6 | 35.0 [c] | 9.8 | 0.9 | 10.4 | 34.4 [b] | |
| | 8 | ND | 7.4 | 73.9 [a] | 9.7 | 1.1 | 9.0 | 38.8 [b] | 0.04 |
| Low Pressure | 4 | ND | 5.4 | 21.9 [c] | 9.6 | 1.1 | 8.7 | 39.7 [b] | |
| | 8 | ND | 6.2 | 55.7 [b] | 9.7 | 1.1 | 9.1 | 42.5 [b] | 0.04 |
| LSD | | NS | 13.9 | NS | NS | NS | NS | 26.3 | NS |
| | After 7 days air storage at 20 °C | | | | | | | | |
| UTC | 4 | ND | 5.4 | 1.6 [d] | 9.8 | 1.0 | 9.4 | 104.8 [b] | |
| | 8 | 12.5 | 6.2 | 3.4 [d] | 9.6 | 1.1 | 8.9 | 159.5 [a] | 0.03 |
| Low Oxygen | 4 | ND | 4.9 | 28.0 [c] | 9.7 | 1.0 | 9.5 | 42.1 [f] | |
| | 8 | 9.1 | 5.0 | 93.5 [a] | 9.6 | 1.0 | 9.3 | 59.8 [cd] | 0.04 |
| Low Pressure | 4 | ND | 5.1 | 21.6 [c] | 9.7 | 1.1 | 9.2 | 56.1 [cd] | |
| | 8 | 9.1 | 4.7 | 67.9 [b] | 9.6 | 1.0 | 9.6 | 72.7 [c] | 0.04 |
| LSD | | NS | 10.2 | NS | NS | NS | NS | 25.6 | NS |

Values with the same superscript letter are not significantly different at Student's *T*-tests LSD using an alpha of 0.05. Each treatment duration analysed separately. Each value is the mean of 3 treatment units. Ethylene and Respiration Rate is the mean of 2 fruit sealed in a respiration jar, TSS is the mean of 10 individual fruits, ethanol and TA are the means of the juice of 10 fruits combined and firmness is the mean of 10 individual fruits (Valencia orange). NB: NS = not significant and ND = not detected <9 µL $C_2H_4$/kg/h.

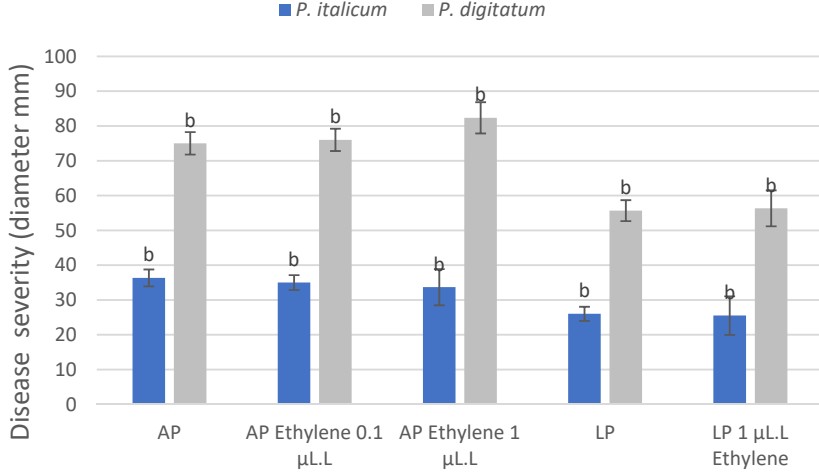

**Figure 4.** Disease severity (mm) of blue mould caused by *Penicillium italicum* and green mould caused by *Penicillium digitatum* in navel oranges following treatment at atmospheric pressure (AP) with 0.1 µL/L, and 1µL/L ethylene and low pressure (LP) with ethylene 1 µL/L. The values are the means of 3 replicates of 15 oranges, with the severity of both moulds analysed separately with Student's t-test using an alpha of 0.05. Mean values with different letters indicate significant differences among pathogens. Vertical lines above columns designate standard deviation. Blue mould severity has an LSD of 3.0057 and green mould severity has an LSD of 9.850.

## 4. Conclusions

The results showed that LP storage of 6.6 kPa and LO treatments of 1% $O_2$ (for 4 and 8 days at 20 °C) decreased *P. digitatum* and *P. italicum* growth in infected oranges. The reduction in blue mould severity observed with the application of these two physical treatments might also potentially be an alternative to chemical fungicides and could contribute to organic or chemical-free citrus production. It was also noted that LP and LO treatments reduced weight loss compared to the UTC. However, these physical treatments resulted in an increase in ethanol levels within the fruit, presumably through anaerobic metabolism, but these increased levels were below levels that the consumer can perceive. The addition of exogenous ethylene at low concentrations was also assessed at atmospheric pressure and low pressure and was found to have no effect on mould development. These experiments support the potential of LP and LO storage for decay reduction in citrus; however, further experiments need to be conducted to test viability to ensure that commercial treatments can be developed. Whilst the introduction of LP may increase storage duration and quality, the concerns in developing treatments on a commercial level include scaling for the LP vessel.

**Supplementary Materials:** The following are available online at https://www.mdpi.com/article/10.3390/horticulturae7120582/s1, Figure S1: Ethylene injection air peak and ethylene level.

**Author Contributions:** Conceptualization, J.A., J.B.G. and P.P.; methodology, J.A., J.B.G., L.P. and P.P.; formal Analysis, J.A. and P.P.; investigation, L.P. and P.P.; resources, J.B.G.; writing—original draft preparation, J.A., J.B.G. and P.P.; writing—review & editing, J.A., J.B.G., P.P., L.P. and Q.V.V.; supervision, J.B.G., P.P., L.P. and Q.V.V.; project administration, J.B.G. All authors have read and agreed to the published version of the manuscript.

**Funding:** This study was funded by the NSW Department of Primary Industries and the University of Newcastle. This project was supported by Horticulture Innovation Australia—'Citrus Postharvest Program' (CT19003).

**Institutional Review Board Statement:** Not applicable.

**Data Availability Statement:** Data available from author on reasonable request.

**Acknowledgments:** This study was funded by the NSW Department of Primary Industries and the University of Newcastle. This project was supported by Horticulture Innovation Australia—'Citrus Postharvest Program' (CT19003).

**Conflicts of Interest:** The authors declare no conflict of interest.

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
