# Peer review of "Effect of Low Pressure and Low Oxygen Treatments on Fruit Quality and the In Vivo Growth of Penicillium digitatum and Penicillium italicum in Oranges"

_horticulturae, doi:10.3390/horticulturae7120582_

Round 1
Reviewer 1 Report
The manuscript entitled " Effect of low pressure and low oxygen treatments on fruit quality and the in vivo growth of Penicillium digitatum and Penicillium italicum in oranges" is a interesting study, well planned and designed, however, there are some concerns, especially in relation to the experimental procedures. The major problems are listed below.
>1 Variety has a large effect on fruit storage life. In this manuscript, two subtypes, Bellamy and Valencia Late, were separately used for physiological and pathological quality tests. Is there any difference between these two subtypes, especially the storage-tolerance of fruit?
>2 The optimum storage temperature for oranges is 2 to 3°C at 85% to 90% relative humidity allowing for storage times of between 4 to 8 weeks. Was there any test about LO on fruit quality and decay at low storage temperature.
>3 L100-101 Inoculated fruit were then incubated for 24 hours at 20°C before treatment started. Please explain why chose 24 hours before treatment started. How about the effect of other time interval on disease severity of fruit.
>4 Table 2. Why the ethylene was ND in table 2. The fruit was not wounded? How was about the production of ethylene in wounded fruit or the infected fruit. Was the ethylene at 1 uL /L was enough to markedly affect the total content of ethylene in the package.
>5 Table 2. The ethanol content (53.9, 73.9, 93.5 ul/L) increases over the test time point (0, 4, 7 d) in LO treatment at 8 d duration. How about the follow-up?
>6 Fig 3. P. digitatum and P. italicum, which one is the dominant pathogen in Navel and Valencia oranges. are their pathogenicity (disease severity) in oranges similar?
>7 Is there no significant difference of disease incidence in all Figures.
>8 All the data in the figures and table lack of standard error or standard deviation.
Author Response
Thank you very much for your feedback, we hope this will address all your concerns.
>1 Variety has a large effect on fruit storage life. In this manuscript, two subtypes, Bellamy and Valencia Late, were separately used for physiological and pathological quality tests. Is there any difference between these two subtypes, especially the storage-tolerance of fruit?
Thank you for the question, Valencia Late are more storage tolerant, but two different varieties were used because they are grown at different times of the year, and we needed to keep working for the entire citrus season to optimize our resources and give confidence in our results. The results from different cultivars give more robust data on potential treatment effects than just one cultivar. Results on decay with artificially inoculated fruit are independent of results on quality of intact healthy fruit, so different cultivars can be used.
>2 The optimum storage temperature for oranges is 2 to 3°C at 85% to 90% relative humidity allowing for storage times of between 4 to 8 weeks. Was there any test about LO on fruit quality and decay at low storage temperature.
No. This experiment was only for shelf life assessments at 20ºC. We will consider assessing the effects on quality of cold-stored fruit in further research.
>3 L100-101 Inoculated fruit were then incubated for 24 hours at 20°C before treatment started. Please explain why chose 24 hours before treatment started. How about the effect of other time interval on disease severity of fruit.
Inoculated fruits were incubated for 24 hours at 20°C after inoculation because we intended to resemble real infections occurring in the field just before or during harvest time. This is the procedure followed by most standardised methods for assessment of antifungal postharvest treatments of citrus fruits using artificial fungal inoculation.
>4 Table 2. Why the ethylene was ND in table 2. The fruit was not wounded? How was about the production of ethylene in wounded fruit or the infected fruit. Was the ethylene at 1 uL /L was enough to markedly affect the total content of ethylene in the package.
The not detected (ND) in table 2 was because the ethylene level was below the level we were able to detect, ie beyond the limit of detection. The fruit used for the quality assessments was not wounded, we suspect wounded fruit would produce larger amounts of ethylene which would be interesting to compare in a future experiment.
>5 Table 2. The ethanol content (53.9, 73.9, 93.5 ul/L) increases over the test time point (0, 4, 7 d) in LO treatment at 8 d duration. How about the follow-up?
It is surprising that the ethanol continues to go up after removal, but these are the measurements we recorded. This may also be interesting to compare in a future experiment.
>6 Fig 3. P. digitatum and P. italicum, which one is the dominant pathogen in Navel and Valencia oranges. are their pathogenicity (disease severity) in oranges similar?
Both are severe citrus postharvest pathogens with similar pathogenicity on all commercial citrus species and cultivars, although at higher temperatures (room temperatures) P. digitatum will grow faster than P. italicum, which is prevalent during long-term cold storage (<5ºC). Actual fruit losses due to each mould in commercial conditions will depend on the epidemiological and fruit handling conditions for each particular situation.
>7 Is there no significant difference of disease incidence in all Figures.
The treatment did not impact disease incidence (Experiment 1 and 4 used 15 fruit and Experiment 2 used 30 fruit) but did affect the disease severity (lesion size, which indicates the growth of the pathogen in the fruit rind).
>8 All the data in the figures and table lack of standard error or standard deviation.
Thank you for the comment, We have added standard deviation.
Reviewer 2 Report
In the manuscript titled, 'Effect of low pressure and low oxygen treatments on fruit quality and the in vivo growth of Penicillium digitatum and Penicillium italicum in oranges', the authors describe a very simple but effective set of experiments to assess the effectiveness of low-pressure and low-oxygen storage on the growth of common fungal pathogens on oranges. Furthermore, they also perform one experiment to assess the effect of these treatments on fruit quality.
I found the overall design of the experiments simple but clever and informative. The results and data were well presented graphically and discussed appropriately in the text. Finally, I found the conclusions to be appropriately stated (not overstated or exaggerated). The writing is clear and concise and holds the reader's attention. Overall one of the best papers I have reviewed this year in terms of writing quality and experimental design and execution.
Author Response
Thank you very much for your time to complet this review
Reviewer 3 Report
The manuscript from Archer et al. describes the effect of low pressure and low oxygen treatments on fruit quality and the influence of Penicillium digitatum and Penicillium italicum growth in oranges. In fact, the article follows the evolution after the intended contamination with the two moulds and the evolution of microbial proliferation in different storage conditions under low pressure and low oxygen. However, it is well known that these microorganisms are aerobic and proliferated in the presence of oxygen. It would be interesting to evaluate in these conditions the proliferation of anaerobic microorganisms (like Clostridium sp.). The manuscript is well written. Thus, it deserves being published after some modifications.
Additional comments/suggestions:
Materials and methods
- Did the harvest take place on the same day? What is the period from harvesting to conducting the experiments? Under what conditions were they stored until the experiments were performed?
- Why did you use another variety of oranges for experiment 3? For a better correlation of the data, the same variety should have been used for all experiments.
- What sources did these fungi come from? Which methods did you use for identification?
- What volume was used for fruits inoculation?
- To determine the ethylene content, what flow had the mobile phase (N2)? What was the run time for data acquisition?
- Check the font, it is not similar in the whole Materials and methods section. Use italic font for Latin names (mould species)
Results and discussion
- Did you perform the microbiological characterization by evaluating the increase / decrease in the temporal dynamics of CFU / mL to highlight the influence of storage conditions on P. italicum contamination? Please reformulate lines 225-226, 250-253.
- For a better evaluation it would be preferable to enter SD (standard deviation) in all graphs and Table 2 (mean ± SD). What are the letters above the bars in the graphs? Mention them in the title of the figures.
- Can you include two chromatograms for additional materials to highlight the lack / presence of ethylene production? Other metabolic gases (e.g. H2S) could also be identified / highlighted (related to the lines 253–254)?
Author Response
Thank you very much for your feedback, we hope this will address all your concerns
Additional comments/suggestions
Materials and methods
- Did the harvest take place on the same day? What is the period from harvesting to conducting the experiments? Under what conditions were they stored until the experiments were performed?
Thank you for your question. We have added the answer to the manuscript. Navel oranges and Valencia oranges were harvested at different times of the year. Fruit were used the same day as harvest or stored up to 1 week at 5 °C and 90% relative humidity (RH)
- Why did you use another variety of oranges for experiment 3? For a better correlation of the data, the same variety should have been used for all experiments.
We conducted experiment 3 with another cultivar because this experiment was conducted later in the season and Bellamy were not available anymore. In any case, results on decay are independent of results on quality.
- What sources did these fungi come from? Which methods did you use for identification?
They were obtained from NSW DPI citrus pathology laboratory and were identified by morphology.
- What volume was used for fruits inoculation?
The fruit were inoculated by dipping a stainless steel rod into the inoculum solution, and then immediately making a puncture 2 mm deep in the fruit. 50 ml of inoculum solution was made up at a time and each orange on average was inoculated with ~50 µL
- To determine the ethylene content, what flow had the mobile phase (N2)? What was the run time for data acquisition?
The mobile phase (N2) was 60ml and the run time was ~15 seconds. We have added this to the manuscript.
- Check the font, it is not similar in the whole Materials and methods section. Use italic font for Latin names (mould species)
Thank you, we have corrected as reviewer suggestions.
Results and discussion
- Did you perform the microbiological characterization by evaluating the increase / decrease in the temporal dynamics of CFU / mL to highlight the influence of storage conditions on P. italicum contamination? Please reformulate lines 225-226, 250-253.
Thank you. Disease development on inoculated fruit was assessed in terms of disease severity (mean diameter of lesions caused by artificial fruit inoculation with a spore suspension of the pathogen). We have changed our statement to say “disease severity”
- For a better evaluation it would be preferable to enter SD (standard deviation) in all graphs and Table 2 (mean ± SD). What are the letters above the bars in the graphs? Mention them in the title of the figures.
Thank you, we have added SD for all Figures, however we believe the data presented as in Table 2 is appropriate for this manuscript.
- Can you include two chromatograms for additional materials to highlight the lack / presence of ethylene production? Other metabolic gases (e.g. H2S) could also be identified / highlighted (related to the lines 253–254)?
We have added two chromatograms as additional materials. We did not identify other metabolic gases but that would be interesting to review in further research.

Round 2
Reviewer 1 Report
none
Author Response
Thank you for your review
Reviewer 3 Report
The authors performed the suggested alterations.
Author Response
thank you for your review